# Why self-attention is Natural for Sequence-to-Sequence Problems? A Perspective from Symmetries

## Abstract

In this paper, we show that structures similar to self-attention are natural to learn many sequence-to-sequence problems from the perspective of symmetry. Inspired by language processing applications, we study the orthogonal equivariance of *seq2seq functions with knowledge*, which are functions taking two inputs—an input sequence and a "knowledge"—and outputting another sequence. The knowledge consists of a set of vectors in the same embedding space as the input sequence, containing the information of the language used to process the input sequence. We show that orthogonal equivariance in the embedding space is natural for seq2seq functions with knowledge, and under such equivariance the function must take the form close to the self-attention. This shows that network structures similar to self-attention are the right structures to represent the target function of many seq2seq problems. The representation can be further refined if a "finite information principle" is considered, or a permutation equivariance holds for the elements of the input sequence.

## 1 Introduction

Neural network models using self-attention, such as Transformers Vaswani et al. (2017), have become the new benchmark in the fields such as natural language processing and protein folding. Though, the design of self-attention is largely heuristic, and theoretical understanding of its success is still lacking. In this paper, we provide a perspective for this problem from the symmetries of sequence-to-sequence (seq2seq) learning problems. By identifying and studying appropriate symmetries for seq2seq problems of practical interest, we demonstrate that structures like self-attention are natural for representing these problems.

Symmetries in the learning problems can inspire the invention of simple and efficient neural network structures. This is because symmetries reduce the complexity of the problems, and a network with matching symmetries can learn the problems more efficiently. For instance, convolutional neural networks (CNNs) have seen great success on vision problems, with the translation invariance/equivariance of the problems being one of the main reasons. This is not only observed in practice, but also justified theoretically Li et al. (2020b). Many other symmetries have been studied and exploited in the design of neural network models. Examples include permutation equivariance Zaheer et al. (2017) and rotational invariance Kim et al. (2020); Chidester et al. (2019), with various applications in learning physical problems. See Section 2.1 for more related works.

In this work, we start from studying the symmetry of seq2seq functions in the embedding space, the space in which each element of the input and output sequences lie. For a language processing problem, for example, words or tokens are usually vectorized by a one-hot embedding using a dictionary. In this process, the order of words in the dictionary should not influence the meaning of input and output sentences. Thus, if a permutation is applied on the dimensions of the embedding space, the input and output sequences should experience the same permutation, without other changes. This implies a permutation equivariance in the embedding space. In our analysis, we consider equivariance under orthogonal group, which is slightly larger than the permutation group. We show that if a function $f$ is orthogonal equivariant in the embedding space, then its output can be expressed as linear combinations of the elements of the input sequence, with the coefficients only depending

on the inner products of these elements. Concretely, let $X \in \mathbb{R}^{d \times n}$ denote an input sequence with length $n$ in the embedding space $\mathbb{R}^d$. If $f(QX) = Qf(X)$ holds for any orthogonal $Q \in \mathbb{R}^{d \times d}$, then there exists a function $g$ such that

$$f(X) = Xg(X^T X).$$

However, the symmetry on the embedding space is actually more complicated than a simple orthogonal equivariance. In Section 3.2, we show that the target function for a simple seq2seq problem is not orthogonal equivariant, because the target function works in a fixed embedding. To accurately catch the symmetry in the embedding space, we propose to study *seq2seq functions with knowledge*, which are functions with two inputs, $f(X, Z)$, where $X \in \mathbb{R}^{d \times n}$ is the input sequence and $Z \in \mathbb{R}^{d \times k}$ is another input representing our "knowledge" of the language. The knowledge lies in the same embedding space as $X$, and is used to extract information from $X$. With this additional input, the symmetry in the embedding space can be formulated as an orthogonal equivariance of $f(X, Z)$, i.e. $f(QX, QZ) = Qf(X, Z)$ for any inputs and orthogonal matrix $Q$. Intuitively understood, in a language application, as long as the knowledge is always in the same embedding as the input sequence, the meaning of the output sequence will not change with the embedding. Based on the earlier theoretical result for simple orthogonal equivariant functions, if a seq2seq function with knowledge is orthogonal equivariant, then it must have the form

$$f(X, Z) = Xg_1(X^T X, Z^T X, Z^T Z) + Zg_2(X^T X, Z^T X, Z^T Z)$$

If $Z$ is understood as a parameter matrix to be learned, the following subset of this representation,

$$f(X, Z) = Xg(X^T Z),$$

is close to a self-attention used in practice, with $Z$ being the concatenation of query and key parameters. This reveals one possible reason behind the success of self-attention based models on language problems.

Based on the results from orthogonal equivariance, we further study the permutation equivariance on the elements of the input sequence. Under this symmetry, we show that seq2seq functions with knowledge have a further reduced form which only involves four different nonlinear functions. Finally, discussions are made on the possible forms of $g$ (or $g_1$ and $g_2$) in the formulations mentioned above. Based on the assumption that these functions are described by a finite amount of information (although their output sizes need to change with respect to the sequence length $n$), we reason that quadratic forms with a nonlinearity used in usual self-attentions is one of the simplest choice of $g$. We also discuss practical considerations that add the complexity of the models used in application compared with theoretical forms.

## 2 BACKGROUND AND RELATED WORK

### 2.1 NEURAL NETWORKS AND SYMMETRIES

Implementing symmetries in neural networks can help the models learn certain problems more efficiently. A well-known example is the success of convolutional neural networks (CNNs) on image problems due to their (approximate) translation invariance LeCun et al. (1989). Many types of symmetries have been explored in the design of neural networks, such as permutation equivariance and invariance Zaheer et al. (2017); Guttenberg et al. (2016); Rahme et al. (2021); Qi et al. (2017a;b), rotational equivariance and invariance Thomas et al. (2018); Shuaibi et al. (2021); Fuchs et al. (2020); Kim et al. (2020), and more Satorras et al. (2021); Wang et al. (2020b); Ling et al. (2016a); Ravanbakhsh et al. (2017). Some works deal with multiple symmetries. In Villar et al. (2021), the forms of functions with various symmetries are studied. These networks see many applications in physical problems, where symmetries are intrinsic in the problems to learn. Examples include fluid dynamics Wang et al. (2020a); Ling et al. (2016b); Li et al. (2020a); Mattheakis et al. (2019), molecular dynamics Anderson et al. (2019); Schütt et al. (2021); Zhang et al. (2018), quantum mechanics Luo et al. (2021a;b); Vieijra et al. (2020), etc. Theoretical studies have also been conducted to show the benefit of preserving symmetry during learning Bietti et al. (2021); Elesedy & Zaidi (2021); Li et al. (2020b); Mei et al. (2021).

### 2.2 SELF-ATTENTION

self-attention Vaswani et al. (2017); Parikh et al. (2016); Paulus et al. (2017); Lin et al. (2017); Shaw et al. (2018) is a type of attention mechanism Bahdanau et al. (2014); Luong et al. (2015) that attends

different elements in a same input sequence. It is the building block of a series of large language models (e.g. Devlin et al. (2018); Brown et al. (2020); Raffel et al. (2020)), and is under extensive research. See Bommasani et al. (2021); Niu et al. (2021) for reviews.

As a preparation for later studies, we briefly summarize the structure of (multihead) self-attention. A self-attention is a seq2seq operator which takes a sequence of vectors as the input, and another sequence of vectors (of the same size) as the output. Let $X \in \mathbb{R}^{d \times n}$ be the input sequence with length $n$. A self-attention computes the output using three parameter matrices: the query parameters $W_Q \in \mathbb{R}^{d_1 \times d}$, the key parameters $W_K \in \mathbb{R}^{d_1 \times d}$, and the value parameters $W_V \in \mathbb{R}^{d \times d}$. Given the input $X$, a "query" and a "key" is computed for every column of $X$ by multiplying with $W_Q$ and $W_K$, i.e. we compute $Q(X) = W_Q X \in \mathbb{R}^{d_1 \times n}$ and $K(X) = W_K X \in \mathbb{R}^{d_1 \times n}$. Then, an attention matrix is obtained by computing the inner product of all pairs of queries and keys:

$$A(X) = Q(X)^T K(X) = X^T W_Q^T W_K X \in \mathbb{R}^{n \times n}.$$

Next, a weight matrix is computed by applying softmax over rows of $A$, and the output of the attention is obtained by a linear combination of the "values", $W_V X$, using rows in the weight matrix as coefficients. In practice, $A(X)$ is usually scaled by a factor of $1/\sqrt{d_1}$, and a residual connection is added, thus we have

$$Attn(X) = X + W_V X \mathcal{S}_r \left( \frac{1}{\sqrt{d_1}} X^T W_Q^T W_K X \right)^T \tag{1}$$

where $\mathcal{S}_r(\cdot)$ computes the softmax of an input matrix over rows.

**Remark 1.** *In this paper we use $X \in \mathbb{R}^{d \times n}$ to denote sequences with length $n$ in the space $\mathbb{R}^d$. Each column of $X$ is an element of the sequence. In many works the same sequence is represented by an $n \times d$ matrix. The two representations are intrinsically equivalent.*

The self-attention mechanism described above consists of one "head", in the sense that we have one query, key and value for each element of $X$. Similar to the way that we add more neurons to a layer of a fully connected neural network, we can add more heads to a self-attention, which gives a multihead attention. For a multihead attention with $m$ heads, we have $m$ different query, key, and value matrices, denoted by $W_Q^{(i)}$, $W_K^{(i)}$, and $W_V^{(i)}$. $W_Q^{(i)}$ and $W_K^{(i)}$ are still $d_1 \times d$ matrices, while $W_V^{(i)}$ are $d_2 \times d$ matrices. Besides, in order to still use the residual connection, an output parameter matrix $W_{out}^{(i)} \in \mathbb{R}^{d \times d_2}$ is added for each head to transform the value vectors in $\mathbb{R}^{d_2}$ into vectors in $\mathbb{R}^d$. With these parameters, each head is similar to a single-head self-attention:

$$head_i(X) = W_V^{(i)} X \mathcal{S}_r \left( \frac{1}{\sqrt{d_1}} X^T (W_Q^{(i)})^T W_K^{(i)} X \right)^T,$$

and the output of the multihead attention is

$$Attn_m(X) = X + \sum_{i=1}^{m} W_{out}^{(i)} head_i(X)$$

$$= X + \sum_{i=1}^{n} W_{out}^{(i)} W_V^{(i)} X \mathcal{S}_r \left( \frac{1}{\sqrt{d_1}} X^T (W_Q^{(i)})^T W_K^{(i)} X \right)^T.$$

**Remark 2.** *In a practical model like a Transformer, a fully-connected layer is sometimes added after multihead attentions. The fully-connected layer is applied to each element of the output sequence.*

## 3 ORTHOGONAL EQUIVARIANCE IN THE EMBEDDING SPACE

In this section, we focus on the orthogonal equivariance in the embedding space. We show that functions with such equivariance enjoy a representation which takes a similar but more general form as self-attention. We start from a theoretical characterization for simple seq2seq functions with orthogonal equivariance (Proposition 1). Then, we introduce and study a class of functions called *seq2seq function with knowledge*, whose form is inspired by typical seq2seq learning problems.

## 3.1 SIMPLE ORTHOGONAL EQUIVARIANT FUNCTIONS

We first consider orthogonal equivariant functions given by the following definition:

**Definition 1.** *Let $\mathcal{X} = \bigcup_{n=1}^{\infty} \mathbb{R}^{d \times n}$ be the space of all sequences in $\mathbb{R}^d$. Let $f : \mathcal{X} \to \mathcal{X}$ be a sequence to sequence function. $f$ is called orthogonal equivariant in the embedding space if for any $X \in \mathcal{X}$ and orthogonal matrix $Q \in \mathbb{R}^{d \times d}$, there is $f(QX) = Qf(X)$.*

For orthogonal equivariant functions, the following proposition shows that any column of the output must be a linear combination of the the columns of $X$, with the coefficients depending only on the inner products between $X$'s columns.

**Proposition 1.** *Let $f : \mathcal{X} \to \mathcal{X}$ be orthogonal equivariant in the embedding space given by Definition 1. Then, there exists a function $g$ taking $X^T X$ as input and producing a matrix with appropriate shape as output, such that for all $X \in \mathcal{X}$, we have*

$$f(X) = Xg(X^T X).$$

Proposition 1 shows that orthogonal equivariant seq2seq functions always represent a linear combination of the elements of their input sequence $X$, with the coefficients being orthogonal invariant. We give the proof in Appendix A. A similar result has appeared in Villar et al. (2021) and played an important role for physics applications.

## 3.2 ORTHOGONAL EQUIVARIANCE WITH "KNOWLEDGE"

Proposition 1 treats seq2seq functions that are strictly orthogonal equivariant in the embedding space. For many practical language problems or other seq2seq learning problems, the embedding indeed has some flexibility over orthogonal transformations–the information is encoded only in the relative positions between vectors in the embedding space, and an orthogonal transformation of those vectors does not change the "meaning" of the sequence, hence the "answer" of the transformed input sequence should be the transformed original answer.

However, this intuitive symmetry does not mean that the target function is orthogonal equivariant. As an example, consider a seq2seq function $f$ that takes an arithmetic expression as the input and outputs the result of the expression, e.g $f(\text{"}2+1\text{"}) = \text{"}3\text{"}$, $f(\text{"}2-1\text{"}) = \text{"}1\text{"}$. The tokens used in the input and output sequences include single digit numbers $0-9$ and arithmetic operators. These tokens can be cast into vectors by an one-hot embedding. To be simple, suppose we only use operators "+" and "-". Then, the embedding space has 12 dimensions. One possible embedding is

$$\text{"}+\text{"} \to e_1, \quad \text{"}-\text{"} \to e_2, \quad \text{"}0\text{"} \to e_3, \quad \text{"}1\text{"} \to e_4, \cdots \text{"}9\text{"} \to e_{12},$$

where $e_i$ is the $i$-th unit vector in the standard orthonormal basis of $\mathbb{R}^{12}$. Under this embedding, $f(\text{"}2+1\text{"}) = \text{"}3\text{"}$ can be written as

$$f([\boldsymbol{e}_5, \boldsymbol{e}_1, \boldsymbol{e}_4]) = [\boldsymbol{e}_6].$$

Now, let $Q_{12} \in \mathbb{R}^{12 \times 12}$ be a linear transformation that swaps the first and second entries of any vector in $\mathbb{R}^{12}$. Then, $Q_{12}$ is orthogonal. If $f$ is orthogonal equivariant, we will have

$$f([\boldsymbol{e}_5, \boldsymbol{e}_2, \boldsymbol{e}_4]) = f(Q[\boldsymbol{e}_5, \boldsymbol{e}_1, \boldsymbol{e}_4]) = [Q\boldsymbol{e}_6] = [\boldsymbol{e}_6].$$

This means $f(\text{"}2-1\text{"}) = \text{"}3\text{"}$, which is obviously not what we expect.

To summarize, the target function is not orthogonal equivariant because it works in a fixed embedding and cannot deal with sequences from different embeddings. The intuitive symmetry we discussed earlier can be understood as the symmetry in an equivalent class of target functions. Let $f$ be a seq2seq function in a certain embedding, if an orthogonal transformation $Q$ is applied to this embedding, then there exists another function $f_Q$ that satisfies

$$f_Q(QX) = Qf(X).$$

$f_Q$ does the same thing as $f$ in a different embedding. Collecting $f_Q$ for all orthogonal transformations $Q$, the set $\{f_Q\}$ is an equivalence class of $f$ in all embeddings (obtained by orthogonal transformations).

The discussion above points out that the target function is aware of the embedding it works in. Intuitively, this is because the function contains some knowledge used to process the input sequence, and the knowledge depends on the embedding. Motivated by this point of view, we propose to study functions which take the "knowledge" as an explicit input. Like the input sequence, the knowledge also consists of vectors in the embedding space, showing its embedding dependence. The knowledge is used to extract information from the input sequence. Concretely, we consider functions $f : \mathcal{X} \times \mathbb{R}^{d \times k} \to \mathcal{X}$ taking two inputs, $X \in \mathcal{X}$ and $Z \in \mathbb{R}^{d \times k}$, with $X$ being the original input sequence, and $Z$ being the knowledge. With this additional knowledge input, the function $f$ can be orthogonal equivariant—changing the embedding transforms $X$ and $Z$ simultaneously, and the true "meaning" of what $f$ does is not changed. In other words, the equivalent class $\{f_Q\}$ is parameterized by the knowledge input such that $f_Q(\cdot) = f(\cdot, QZ)$.

From now on, we study *orthogonal equivariant functions with knowledge*, whose definition is given below:

**Definition 2.** *Let $f : \mathcal{X} \times \mathbb{R}^{d \times k} \to \mathcal{X}$ be a seq2seq function with knowledge. For any $Z \in \mathbb{R}^{d \times k}$, $f$ is called orthogonal equivariant with knowledge $Z$ if for any $X \in \mathcal{X}$ and orthogonal matrix $Q \in \mathbb{R}^{d \times d}$, there is $f(QX, QZ) = Qf(X, Z)$.*

As a corollary of Proposition 1, we have the following proposition characterizing the formulation of functions satisfying Definition 2.

**Proposition 2.** *Let $Z \in \mathbb{R}^{d \times k}$, and $f : \mathcal{X} \times \mathbb{R}^{d \times k} \to \mathcal{X}$ be a function that is orthogonal equivariant with knowledge $Z$. Then, there exist two functions $g_1$ and $g_2$ independent of $Z$, taking $X^T X, Z^T X, Z^T Z$ as inputs, and producing matrices with appropriate shapes as outputs, such that for all $X \in \mathcal{X}$, we have*

$$f(X, Z) = X g_1(X^T X, Z^T X, Z^T Z) + Z g_2(X^T X, Z^T X, Z^T Z). \tag{2}$$

*Proof.* Let $\tilde{X} = [X, Z] \in \mathbb{R}^{d \times (n+k)}$. Viewed as a function of $\tilde{X}$, $f$ satisfies $f(Q\tilde{X}) = Qf(\tilde{X})$ for any orthogonal matrix $Q \in \mathbb{R}^{d \times d}$. Hence, by Proposition 1, there exists a function $g$ depending on $\tilde{X}^T \tilde{X}$, such that

$$f(\tilde{X}) = \tilde{X} g(\tilde{X}^T \tilde{X}).$$

By the definition of $\tilde{X}$, $g$ can be written as a function of $X^T X$, $Z^T X$ and $Z^T Z$, i.e. $g(\tilde{X}^T \tilde{X}) = g(X^T X, Z^T X, Z^T Z)$. Noticing that $\tilde{X}$ has $n + k$ columns, $g$ must have $n + k$ rows. Letting

$$g(X^T X, Z^T X, Z^T Z) = \left[ \begin{array}{c} g_1(X^T X, Z^T X, Z^T Z) \\ g_2(X^T X, Z^T X, Z^T Z) \end{array} \right]$$

with $g_1$ taking the first $n$ rows and $g_2$ taking the next $k$ rows, we have

$$f(X, Z) = X g_1(X^T X, Z^T X, Z^T Z) + Z g_2(X^T X, Z^T X, Z^T Z).$$

$\square$

In practice, the knowledge $Z$ in a function $f(X, Z)$ studied above can be treated as a parameter matrix learned during the training process. We note that the self-attention in equation 1 takes a similar form. In the self-attention, the product of $X$ with the attention matrix has the form $X g(Z^T X)$, with $Z = [W_Q^T, W_K^T]$, and $g$ being the composition of a quadratic function and a softmax operation:

$$g(Y) = \mathcal{S}_r \left( \frac{1}{\sqrt{d_1}} Y^T \left[ \begin{array}{cc} 0 & I \\ 0 & 0 \end{array} \right] Y \right).$$

Indeed, similar as our understanding of $Z$, the query and key parameters in the self-attention are usually understood as knowledge of the language used to extract information from the input sequence. These parameters are naturally embedding dependent. Certainly, the self-attention used in practice contains more components than merely a $X g(Z^T X)$ form. For example, as shown in equation 1, a linear transformation in the embedding space is applied by $W_V$, and a residual connection is added. in Section 5, we discuss some practical considerations that may cause additional complication of the model in practice.

Coming back to the formulation 2, if $Z$ is understood as a parameter matrix, it is fixed after training. Then, among the three inputs of $g_1$ and $g_2$, $Z^T Z$ is a constant, and $X^T X$ is an identity matrix under

one-hot embedding. Hence, $Z^T X$ is the most informative input. Moreover, since $Z$ is a constant matrix, the linear combination of its columns, $Z g_2(X^T X, Z^T X, Z^T Z)$ becomes less important than the linear combination of $X$'s columns, $X g_1(X^T X, Z^T X, Z^T Z)$. Extracting the most meaningful parts in the formulation 2, we obtain a simpler form $f(X, Z) = X g_1(Z^T X)$. This coincides with what appears in the self-attention.

### 3.3 Finite information and the representation of coefficients

In formulation 2 or the simplified formulation $f(X, Z) = X g_1(Z^T X)$, the coefficient functions $g_1$, $g_2$ can be quite arbitrary. They can have very complicated dependence with their inputs. For example, for a function $f(X, Z) = X g(Z^T X)$ whose output has the same length as the input, when $X \in \mathbb{R}^{d \times n}$, we have $g(Z^T X) \in \mathbb{R}^{n \times n}$. In the most general case, $g$ can have a different formulation $g_n : \mathbb{R}^{k \times n} \to \mathbb{R}^{n \times n}$ for each $n$. Because $n$ can be arbitrarily large, the description of $g$ requires infinite amount of information. However, if these functions can be described and implemented by machine learning models, they must contain only a finite amount of information. In other words, the functions cannot get infinitely complicated when the sizes of their inputs become large. In this section, based on this *finite information principle*, we discuss possible forms of the $g$'s.

For the convenience of the discussion, we focus on the form $f(X, Z) = X g(Z^T X)$ and assume that the output of $f$ has the same length as its input. Hence, for any input $X \in \mathbb{R}^{d \times n}$, we have $g(Z^T X) \in \mathbb{R}^{n \times n}$. In this case, we put our discussion on $g$ under the following more specific statement of the finite information principle:

**Assumption 1.** *(Finite information principle) $g$ is represented by a parameterized model with a finite number of parameters not depending on $n$.*

This assumption concerns only one aspect of the broader idea of "finite information". But it is the only aspect that we can quantify easily.

Now, we consider parameterized representations for $g$. Given Assumption 1, one of the simplest parameterizations is the composition of a nonlinear function and a quadratic forms, such as $\sigma(X^T Z A Z^T X)$ for some matrix $A \in \mathbb{R}^{k \times k}$. To see this, denote $Y = Z^T X \in \mathbb{R}^{k \times n}$ and consider $g$ represented by a composition of an elementwise nonlinear function and a sum of matrix products involving $Y$, i.e.

$$g(Y) = \sigma \left( \sum_{i=1}^{N} W_{i,0} \prod_{j=1}^{K_i} \tilde{Y} W_{i,j} \right), \tag{3}$$

where $\tilde{Y}$ is either $Y$ or $Y^T$, and $N$ can be infinity. In the formulation above, $W_{i,j}$ are parameters. By the finite information principle, the dimensions of $W_{ij}$ in equation 3 should not depend on $n$. Then, it is easy to show that we always have $K_i \geq 2$ in equation 3, because terms with $K_i = 0$ or 1 cannot have shape $n \times n$ without $n$-dependent parameter matrices. Hence, there is no constant or linear terms in the sum of matrix products, and thus the simplest terms are quadratic terms. In its simplest form, without higher order terms, we have $g(Y) = \sigma(Y^T W Y)$ for some $W \in \mathbb{R}^{k \times k}$, in which case the output always has the shape $n \times n$ for any $n$. Note that the self-attention matrix used in practice is very close to this form. If $Z$ is the concatenation of the query and key matrices, i.e. $Z = [W_Q^T, W_K^T]$, then by taking $A = \begin{bmatrix} 0 & I \\ 0 & 0 \end{bmatrix}$ we have

$$X^T Z A Z^T X = X^T W_Q^T W_K X.$$

The only difference is that the softmax operation is not elementwise.

### A perspective from kernels

Another perspective to create $g$ with finite amount of information is from the kernels. Viewing the input $Y \in \mathbb{R}^{k \times n}$ as $n$ vectors in $\mathbb{R}^k$, $g$ maps the $n$ vectors into an $n \times n$ matrix, characterizing the relations between these vectors. This can naturally be achieved by a kernel function $K(\cdot, \cdot) : \mathbb{R}^k \times \mathbb{R}^k \to \mathbb{R}$. Denote $Y = [\boldsymbol{y}_1, ..., \boldsymbol{y}_n]$, then we can let $g(Y) = (K(\boldsymbol{y}_i, \boldsymbol{y}_j))_{n \times n}$. When $K$ is an inner product kernel $K(\mathbf{x}, \boldsymbol{y}) = \sigma(\mathbf{x}^T \boldsymbol{y})$, which is widely used in traditional machine learning models such as the support vector machine, $g$ takes a similar quadratic form (with an elementwise

nonlinearity) as in the discussion above, i.e. $g(Y) = \sigma(Y^T Y)$. Besides, there are more kernels to choose. For instance, a radio basis function (RBF) kernel $K(\mathbf{x}, \boldsymbol{y}) = f(\|\mathbf{x} - \boldsymbol{y}\|)$ can produce a $g$ defined by $g_{ij}(Y) = f(\|\boldsymbol{y}_i - \boldsymbol{y}_j\|)$. These representations of coefficients may see benefits in some special applications. Actually, self-attention using kernels has already been studied in previous works such as Rymarczyk et al. (2021); Chen et al. (2021)

## 4 PERMUTATION EQUIVARIANCE FOR SEQUENCE ELEMENTS

in this section, we consider another symmetry—the permutation equivariance for the elements of the sequence. With this permutation equivariance, the form 2 can be further restricted. In a seq2seq problem such as a language problem, though, the order of the input is usually important. Hence, permutation equivariance on the order of the sequence should not be expected. However, in practice some parts of the problems or models may have permutation equivariance. For example, when self-attention based models are used to learn seq2seq problems, a position encoding is usually added to the input sequence before fed into the model Vaswani et al. (2017). In this case, the order information is included in the input sequence and the function implemented by the model can be permutation equivariant.

For any sequence $X \in \mathbb{R}^{d \times n}$, we call $n$ the length of $X$, denoted by $l(X)$. We consider the following definition of permutation equivariance:

**Definition 3.** *Let $f : \mathcal{X} \times \mathbb{R}^{d \times k} \to \mathcal{X}$ be a seq2seq function with knowledge. Assume $l(f(X, Z)) = l(X)$ always holds. For any $Z \in \mathbb{R}^{d \times k}$, $f$ is called elementwise permutation equivariant with knowledge $Z$ if for any permutation matrix $P \in \mathbb{R}^{l(X) \times l(X)}$, we have $f(XP, Z) = f(X, Z)P$.*

Based on the discussions in previous sections, we focus on functions with the form $f(X, Z) = Xg(Z^T X)$. Given the additional permutation equivariance in Definition 3, we have the following proposition that further narrows down the form of the functions. The proof of the proposition is given in Appendix B.

**Proposition 3.** *Let $f$ be a function with form $f(X, Z) = Xg(Z^T X)$. Assume $f$ is elementwise permutation equivariant with knowledge $Z$. Then, for any specific $n$, there exist functions $\rho_1$, $\rho_2$, $\psi_1$, $\psi_2$, such that for any $X \in \mathbb{R}^{d \times n}$ with full column rank, we have*

$$g_{ii}(Z^T X) = \rho_1\big(\mathbf{x}_i, Z, \sum_{k=1, \; k \neq i}^{n} \psi_1(\mathbf{x}_k; \mathbf{x}_i, Z)\big)$$

$$g_{ij}(Z^T X) = \rho_2\big(\mathbf{x}_i, \mathbf{x}_j, Z, \sum_{k=1, \; k \neq i,j}^{n} \psi_2(\mathbf{x}_k; \mathbf{x}_i, \mathbf{x}_j, Z)\big)$$

*for $i, j = 1, 2, ..., n$ and $j \neq i$. Here, $g_{ij}$ are the $(i, j)$-th component function of $g$, i.e. $g = (g_{ij})_{n \times n}$.*

**Remark 3.** *For a self-attention layer used in practice, $Z = [W_Q, W_K]$, in which case we have*

$$g_{ij}(Z^T X) = \frac{e^{\mathbf{x}_i^T W_Q^T W_K \mathbf{x}_j}}{\sum_{k=1}^{n} e^{\mathbf{x}_i^T W_Q^T W_K \mathbf{x}_k}}.$$

*Using the form in Proposotion 3, this $g$ can be obtained by taking*

$$\psi_2(\mathbf{x}; \boldsymbol{y}, \boldsymbol{z}, Z) = e^{\boldsymbol{y}^T W_Q^T W_K \mathbf{x}}, \quad \rho_2(\mathbf{x}, \boldsymbol{y}, Z, \psi) = \frac{e^{\mathbf{x}^T W_Q^T W_K \boldsymbol{y}}}{e^{\mathbf{x}^T W_Q^T W_K \boldsymbol{y}} + \psi},$$

*and taking*

$$\psi_1(\mathbf{x}; \boldsymbol{y}, Z) = \psi_2(\mathbf{x}, \boldsymbol{y}, \boldsymbol{y}, Z), \quad \rho_1(\mathbf{x}, Z, \psi) = \rho_2(\mathbf{x}, \mathbf{x}, Z, \psi).$$

## 5 PRACTICAL CONSIDERATIONS

In previous sections, we revealed the natural forms of seq2seq functions that satisfy specific symmetries that are reasonable for many practical problems. Therefore, the structures identified can

be taken into consideration when designing neural network models to learn these problems, as approaches to improve the efficiency of learning. The self-attention, although designed without utilizing these connections between symmetries and structures, has structures that coincides with the forms we identified. This may partially explain the success of self-attention based models.

Usually, the models used in practice have to be more complicated than that given by the theory, to address practical issues that are not caught in the simplified setting of the theory. For CNNs, for example, convolution layers are stacked to extract features hierarchically, and normalization layers are added to help training. In the following, we discuss several considerations when the theories built in the previous sections are used in practical applications.

### THE EVOLUTION OF EMBEDDINGS

In our analysis for orthogonal equivariant functions, we assume the output and the input of the functions are in the same embedding. In practice this might not be true. For example, for a translation problem the input and the output sequences are in two different languages, and hence they may not share one embedding. In this case, we need to implement a mechanism to change the embedding of the output sequence. The simplest way is to apply an elementwise linear transformation to the output, i.e. for a function $f$ with form $f(X, Z) = Xg(Z^T X)$, we can build a new function $\tilde{f}$ by multiplying a matrix on the left of the output of $f$:

$$\tilde{f}(X, Z, W) = WXg(Z^T X). \tag{4}$$

A more flexible way is to apply a general elementwise nonlinear transformation to the output, which can be achieved for example by a two-layer neural network, as used in many self-attention based models:

$$\tilde{f}(X, Z, U, V) = V\sigma\big(UXg(Z^T X)\big), \tag{5}$$

where $U, V$ are parameter matrices, and $\sigma$ is an elementwise nonlinear activation function.

### HIGHER CAPACITY

In the application of neural networks, higher capacity of the model is desired in many cases. Giving the model more flexibility compared to the theoretical formulation can help improve the performance of the model, as long as the flexibility does not impair the training efficiency. Based on the structures in equation 4 or 5, more flexibility can be added to the model by considering a "multihead" version of such functions. For example, a multihead version for 4 with $m$ heads can be

$$\tilde{f}_m(X, Z, W) = \sum_{i=1}^{m} W_i Xg_i(Z_i^T X), \tag{6}$$

where $Z_1, ..., Z_m$ and $W_1, ..., W_m$ are different matrices, and $Z = [Z_1, ..., Z_m]$, $W = [W_1, ..., W_m]$, and $g_1, ..., g_m$ are different functions. This structure is similar to the multihead self-attention.

### COMPOSITIONS AND HIERARCHICAL FEATURE EXTRACTION

A very successful way to increase the capacity of a model and improve the performance of learning is to stack several modules compositionally to form a deep model. A deep model with many layers can extract the information from its input hierarchically. This is the intuitive reason behind the success of deep neural networks. For sequence to sequence applications, we can also stack structures like 6 into a deep model. For example, a model with $L$ layers can be

$$h^{(0)} = X; \quad h^{(l)} = \sum_{i=1}^{m} W_i^{(l)} h^{(l-1)} g((Z_i^{(l)})^T h^{(l-1)}, ), 1 \le l \le L; \quad f(X, Z, W) = h^{(L)},$$

where $Z$ and $W$ include all $Z_i^{(l)}$ and $W_i^{(l)}$ parameters, respectively. This structure looks similar to the successful large language models used in practice. One difference is that a residual link is added on each layer of those models to help the training. Another difference is that elementwise fully connected layers are added after some self-attentions, which can be understood as stacking structures in equation 5.

## 6 SUMMARY

In this paper, we study the representations of sequence-to-sequence functions with certain symmetries, and show that such functions have forms similar to the self-attention. Hence, self-attention seems to be the natural structure to learn many seq2seq problems. Moreover, except the inner product based attention mechanism widely used nowadays, our study reveals more possibilities that may be picked in the design of attention mechanisms, such as higher-order matrix products or the RBF kernels. These forms arise from the discussion on the finite information principle. As a limitation, our discussion on the forms of $g$ in Section 3.3 started from a simple general form 3. More general discussions and more precise characterizations of the finite information principle is left as an important future work.

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

## A  PROOF OF PROPOSITION 1

*Proof.* Consider $X = [\mathbf{x}_1, \mathbf{x}_2, ..., \mathbf{x}_n] \in \mathbb{R}^{d \times n} \subset \mathcal{X}$. We first show that the columns of $f(X)$ lie in the span of $\mathbf{x}_1, ..., \mathbf{x}_n$. Without loss of generality, we assume $f(X)$ has only one column, i.e. $f(X) \in \mathbb{R}^d$. Let $V = \text{span}(\mathbf{x}_1, ..., \mathbf{x}_n)$. Then, there exist $\boldsymbol{v} \in V$ and $\boldsymbol{u} \in V^\perp$, such that $f(X) = \boldsymbol{v} + \boldsymbol{u}$. Let $Q_{\boldsymbol{u}}$ be the Householder transformation

$$Q_{\boldsymbol{u}} = I - \frac{2}{\|\boldsymbol{u}\|^2} \boldsymbol{u}\boldsymbol{u}^T.$$

Then, $Q_{\boldsymbol{u}}$ is an orthogonal matrix. By its definition, we have $Q_{\boldsymbol{u}}\boldsymbol{u} = -\boldsymbol{u}$, and $Q_{\boldsymbol{u}}\boldsymbol{w} = \boldsymbol{w}$ for any $\boldsymbol{w} \perp \boldsymbol{u}$, which implies $Q_{\boldsymbol{u}}\boldsymbol{v} = \boldsymbol{v}$ and $Q_{\boldsymbol{u}}\mathbf{x}_i = \mathbf{x}_i$ for all $i = 1, 2, ..., n$. Since $f$ is orthogonal equivariant, we have
$$f(Q_{\boldsymbol{u}}X) = Q_{\boldsymbol{u}}f(X) = Q_{\boldsymbol{u}}(\boldsymbol{v} + \boldsymbol{u}) = \boldsymbol{v} - \boldsymbol{u}.$$

On the other hand, since $Q_{\boldsymbol{u}}X = X$, we must have

$$f(Q_{\boldsymbol{u}}X) = f(X) = \boldsymbol{v} + \boldsymbol{u}.$$

Therefore, we have $\boldsymbol{u} = 0$, and $f(X) = \boldsymbol{v} \in V$.

Next, we show that the coefficient of the linear combinations can be taken as orthogonal invariant functions. By the analysis above, there exists a function $g$ of input $X$, such that

$$f(X) = Xg(X).$$

The size of $g$'s output depends on $X$ and $f(X)$. Because $f$ is orthogonal equivariant, for any orthogonal matrix $Q \in \mathbb{R}^{d \times d}$ we have

$$QXg(QX) = f(QX) = Qf(X) = QXg(X),$$

which means $Xg(QX) = Xg(X)$. Obviously, we can choose $g$ to satisfy $g(QX) = g(X)$ for any orthogonal $Q$.

Finally, we invoke the first fundamental theorem of invariant theory for the orthogonal group Weyl (1946); Procesi (2006), which states that $g$ only depends on $X$ via $X^T X$. This completes the proof. $\qquad\square$

## B  PROOF OF PROPOSITION 3

*Proof.* With an abuse of notations, we use $g(X, Z)$ to denote the output of $g$ given inputs $X$ and $Z$, despite that $g$ only depends on $Z^T X$. Since $X$ has full column rank, we have $X^\dagger X = I$. Hence, $g(X, Z) = X^\dagger f(X, Z)$. By Definition 3, for any permutation matrix $P \in \mathbb{R}^{n \times n}$, we have

$$g(XP, Z) = P^T X^\dagger f(XP, Z) = P^T X^\dagger f(X, Z)P = P^T g(X, Z)P. \tag{7}$$

Therefore, applying any permutation on $X$ leads to the same permutation on the rows and column of $g(X, Z)$. Recall that the $(i, j)$-th entry of $g$ is given by the function $g_{ij}$. Denote the output of $g_{ij}$ given input $X$ and $Z$ by $g_{ij}(\mathbf{x}_1, ..., \mathbf{x}_n, Z)$. We then study the forms of $g_{ij}$ using equation 7.

First, consider a permutation $P_{1i}$ that swaps $\mathbf{x}_1$ and $\mathbf{x}_i$. By equation 7, we have $g(XP_{1i}, Z)_{11} = g(X, Z)_{ii}$, which means

$$g_{ii}(\mathbf{x}_1, \cdots, \mathbf{x}_i, \cdots, \mathbf{x}_n, Z) = g_{11}(\mathbf{x}_i, \cdots, \mathbf{x}_1, \cdots, \mathbf{x}_n, Z).$$

Hence, all $g_{ii}$ can be generated by $g_{11}$ with a swap permutation of its inputs. For $g_{11}$, if we apply a permutation that is identity on 1, the output of $g_{11}$ does not change although the order of inputs is changed. This means $g_{11}$ is permutation invariant with the inputs $\mathbf{x}_2, ..., \mathbf{x}_n$. By Theorem 2 in Zaheer et al. (2017), viewed as a function of $\mathbf{x}_2, ..., \mathbf{x}_n$, $g_{11}$ has the form $\rho(\sum_{k=2}^n \psi(\mathbf{x}_k))$ for some functions $\rho$ and $\psi$. Considering the inputs $\mathbf{x}_1$ and $Z$, the functions $\rho$ and $\psi$ above depend on $\mathbf{x}_1$ and $Z$. Therefore, there exist functions $\rho_1$ and $\psi_1$, such that

$$g_{11}(X, Z) = \rho_1(\mathbf{x}_1, Z, \sum_{k=2}^n \psi_1(\mathbf{x}_k; \mathbf{x}_1, Z)).$$

By the relation between $g_{11}$ and $g_{ii}$, we have

$$g_{ii}(X, Z) = \rho_1(\mathbf{x}_i, Z, \sum_{k \neq i} \psi_1(\mathbf{x}_k; \mathbf{x}_i, Z)).$$

for any $i = 1, 2, ..., n$.

Next, we consider $g_{ij}$ with $i \neq j$. Without loss of generality, assume $i < j$. Let $P_{1i, 2j}$ be a permutation that swaps $\mathbf{x}_1$ with $\mathbf{x}_i$, and $\mathbf{x}_2$ with $\mathbf{x}_j$. By the permutation equivariance, we have

$$g_{ij}(\mathbf{x}_1, \mathbf{x}_2, \cdots, \mathbf{x}_i, \cdots, \mathbf{x}_j, \cdots, \mathbf{x}_n, Z) = g_{12}(\mathbf{x}_i, \mathbf{x}_j, \cdots, \mathbf{x}_1, \cdots, \mathbf{x}_2, \cdots, \mathbf{x}_n, Z),$$

which means any $g_{ij}$ with $i \neq j$ can be generated by $g_{12}$. Focusing on $g_{12}$, similar to the arguments for $g_{11}$, it is easy to show that $g_{12}$ is permutation invariant with inputs $\mathbf{x}_3, ..., \mathbf{x}_n$. Therefore, there exist functions $\rho_2$ and $\psi_2$, such that

$$g_{12}(X, Z) = \rho_2(\mathbf{x}_1, \mathbf{x}_2, Z, \sum_{k=3}^{\infty} \psi_2(\mathbf{x}_k; \mathbf{x}_1, \mathbf{x}_2, Z)).$$

Hence,

$$g_{ij}(X, Z) = \rho_2(\mathbf{x}_i, \mathbf{x}_j, Z, \sum_{k \neq i,j} \psi_2(\mathbf{x}_k; \mathbf{x}_i, \mathbf{x}_j, Z)).$$

$\square$

