# OpenReview forum: "Why Self Attention is Natural for Sequence-to-Sequence Problems? A Perspective from Symmetries"
_ICLR.cc/2023/Conference — Submitted to ICLR 2023_

### Official Review · Reviewer_KVpw · 2022-10-24

**Confidence:** 3
**Correctness:** 3
**Technical Novelty And Significance:** 4
**Empirical Novelty And Significance:** Not applicable
**Recommendation:** 5

**Clarity, Quality, Novelty And Reproducibility:**

The paper delivers the hypothesis that the success of self-attention based language model, is owing to the fact that self-attention is close to “seq2seq function with knowledge”,  a function class that is guaranteed to be orthogonal equivariant. This idea is very interesting and novel to me.

The illustrations are also clear and convincing to me.

**Strength And Weaknesses:**

Strength:
The paper reveals that one possible reason behind the success of self-attention based language model, is because self-attention is close to “seq2seq function with knowledge”,  a function class that is guaranteed to be orthogonal equivariant. The paper shows that functions with such equivariance enjoy a representation which takes a similar but more general form as self-attention. This idea is fresh and novel to me. I enjoy the clear illustration of the math examples supporting the authors main idea much.

Weakness:
However, I guess the main weakness of the paper is the lack of empirical study. The paper seems does not provide any empirical evidence supporting the hypothesis present in the paper. Take for example, is the seq2seq function with knowledge with equivariance performs better than the conventional self-attention? Is it possible to verify the superiority of the class of function with equivariance empirically? I sincerely appreciate it if authors may kindly comment on the missing empirical study during rebuttal.


**Summary Of The Paper:**

This paper introduces and studies a class of functions called seq2seq function with knowledge, whose form is inspired by typical seq2seq learning problems . This new class of functions explains why  network structures similar to self-attention are the right structures to represent the target function of many seq2seq problems. The paper introduce several concrete illustration to verify the hypothesis proposed in this paper.

**Summary Of The Review:**

The idea present in the paper, i,e., the success of self-attention based language model, is because self-attention is close to “seq2seq function with knowledge”,  a function class that is guaranteed to be orthogonal equivariant is interesting and novel to me.

However, I am concerned that there is no empirical evidence supporting the hypothesis proposed in the paper. I think, w.r.t. the content present in this paper, empirical evidence might be a valid justification supporting the hypothesis in the paper.

---

### Official Review · Reviewer_AZo2 · 2022-10-25

**Confidence:** 1
**Correctness:** 3
**Technical Novelty And Significance:** 3
**Empirical Novelty And Significance:** Not applicable
**Recommendation:** 6

**Clarity, Quality, Novelty And Reproducibility:**

The paper is clearly written and is novel, giving a theoretical analysis on self-attention.

**Strength And Weaknesses:**

The paper introduces an interesting idea that orthogonal equivariance in the embedding space is natural for seq2seq functions with knowledge, and under such quivariance the function must take the form close to the self-attention. It also gives a thorough derivation on the findings. I like this part.

However, my main concern is that the paper is lack of empirical studies on the findings. For example, their theoretical study shows that there could be other design of self-attention, such as high-order matrix products or RBF kernels. It would be interesting to see some experimental results on these.

**Summary Of The Paper:**

The paper studies the representations of seq2seq functions with certain symmetries, and show that such functions have forms similar to the self-attention.

**Summary Of The Review:**

This is a nice theoretical paper but lack of empirical study.

---

### Official Review · Reviewer_KdaA · 2022-10-26

**Confidence:** 4
**Correctness:** 3
**Technical Novelty And Significance:** 2
**Empirical Novelty And Significance:** 3
**Recommendation:** 3

**Clarity, Quality, Novelty And Reproducibility:**

Clarity: The paper is presented in a relatively mathematical dense manner.

Quality & Novelty: The main novelty lies in the perspective of analyzing the success of self-attention from symmetry.

Reproducibility: The paper is a pure theoretical analysis paper.


**Strength And Weaknesses:**

Strength:

+ Study the representations of sequence-to-sequence functions with certain symmetries, and show that such functions have forms similar to the self-attention. Hence, self-attention seems to be the natural structure to learn many seq2seq problems.

+ Moreover, except the inner product based attention mechanism widely used nowadays, the paper revealed more possibilities that may be picked in the design of attention mechanisms, such as higher-order matrix products or the RBF kernels.

Weaknesses:

- The paper only provided pure theoretical analysis. Its real-world relevance should be justified through different Seq2Seq problems.

- Many of the analysis and derivations seem to be not very straightfoward and loss connection to some extent. The paper provides the analysis of symmetry from orthogonal and permuation while there are lots of other possibility of symmetry.

- The paper did not provide an in-depth discussions on the limitations.

**Summary Of The Paper:**

The paper showed that structures similar to self-attention are natural to learn many sequence-to-sequence problems from the perspective of symmetry. It studied the representations of sequence-to-sequence functions with certain symmetries, and showed that such functions have forms similar to the self-attention. Hence, self-attention seems to be the natural structure to learn many seq2seq problems.


**Summary Of The Review:**

The paper showed that structures similar to self-attention are natural to learn many sequence-to-sequence problems from the perspective of symmetry. The main limitaitons lie in 1) theoretical justification; 2) experimental validations.

---

### Decision · Program_Chairs · 2023-01-20

**Decision:**

Reject

**Justification For Why Not Higher Score:**

There are only theoretical results. It is not clear what the implication of the work is.

**Justification For Why Not Lower Score:**

As described above.

**Metareview: Summary, Strengths And Weaknesses:**

Strength
* Theoretical analysis is conducted on self-attention from the perspective of symmetries. It shows that self-attention is close to Seq2Seq function with knowledge, a function class that is guaranteed to be orthogonal equivariant.
* There are interesting theoretical results, such as that there could be other designs of self-attention, such as high-order matrix products or RBF kernels.
* The idea is novel. Readers can get some inspiration from the paper.

Weakness
* The paper provides a purely theoretical analysis. Its practical justification should be given through real Seq2Seq problems.
* The presentation can be further improved. Many of the analyses and derivations are not very straightforward.
* There is no empirical study. It isn't easy to judge the significance of the work.


**Summary Of Ac-Reviewer Meeting:**

I initiated a discussion.  There was no reply from the reviewers.